# Direct binding of TFEα opens DNA binding cleft of RNA polymerase

Sung-Hoon Jun [1,2,3,6✉], Jaekyung Hyun [1,4,6], Jeong Seok Cha [2], Hoyoung Kim[2], Michael S. Bartlett[5], Hyun-Soo Cho [2,7✉] & Katsuhiko S. Murakami [3,7✉]

Opening of the DNA binding cleft of cellular RNA polymerase (RNAP) is necessary for transcription initiation but the underlying molecular mechanism is not known. Here, we report on the cryo-electron microscopy structures of the RNAP, RNAP-TFEα binary, and RNAP-TFEα-promoter DNA ternary complexes from archaea, *Thermococcus kodakarensis* (*Tko*). The structures reveal that TFEα bridges the RNAP clamp and stalk domains to open the DNA binding cleft. Positioning of promoter DNA into the cleft closes it while maintaining the TFEα interactions with the RNAP mobile modules. The structures and photo-crosslinking results also suggest that the conserved aromatic residue in the extended winged-helix domain of TFEα interacts with promoter DNA to stabilize the transcription bubble. This study provides a structural basis for the functions of TFEα and elucidates the mechanism by which the DNA binding cleft is opened during transcription initiation in the stalk-containing RNAPs, including archaeal and eukaryotic RNAPs.

[1] Electron Microscopy Research Center, Korea Basic Science Institute, Chungcheongbukdo 28119, Republic of Korea. [2] Department of Systems Biology, College of Life Science and Biotechnology, Yonsei University, Seoul 03722, Republic of Korea. [3] Department of Biochemistry and Molecular Biology, The Pennsylvania State University, University Park, PA 16802, USA. [4] Molecular Cryo-electron Microscopy Unit, Okinawa Institute of Science and Technology Graduate University, Okinawa 904-0495, Japan. [5] Department of Biology, the Portland State University, Portland, OR 97207, USA. [6] These authors contributed equally: Sung-Hoon Jun, Jaekyung Hyun. [7] These authors jointly supervised this work: Sung-Hoon Jun, Hyun-Soo Cho, Katsuhiko S. Murakami. ✉email: jsh100@kbsi.re.kr; hscho8@yonsei.ac.kr; kum14@psu.edu

Transcription is the first step of gene expression, performed by cellular RNA polymerases (RNAPs) in the three life domains bacteria, archaea, and eukaryote. The structures of RNAP are highly conserved in archaea and eukaryotes, and consist of a "crab-claw" shape core with a protruding stalk domain[1–3].

For promoter-specific transcription, RNAP and general transcription factors (GTFs) are recruited to the promoter DNA to form the pre−initiation complex (PIC)[4]. The melting of DNA around the transcription start site (TSS) converts PIC to the open complex (OC), which is competent to initiate RNA synthesis. Archaeal RNAP requires three general transcription factors (GTFs)––TBP, TFB, and TFEα[5]––whereas Pol II requires more complex GTFs including TFIIA, TFIIB, TFIID/TBP, TFIIE, TFIIF, TFIIH, and the Mediator complex[4,6]. The orthologues of archaeal TBP, TFB, and TFEα are TBP, TFIIB, and TFIIE α-subunit, respectively, which indicates that the archaeal transcription system requires a subset of Pol II GTFs. Due to its simplicity, the archaeal transcription machinery is considered to be the evolutionary precursor and simplified form of the Pol II transcription apparatus that retains the conserved functions between two systems.

The DNA binding cleft of cellular RNAPs can adopt open or closed state by the conformational changes of the clamp domain of RNAP. Opening of the DNA binding cleft is required for promoter DNA loading into the cleft[6]. For Pol II, determination of PIC and OC structures was achieved in yeast and human[7–9]. Yeast PIC structures show the closed cleft, whereas it shows the open cleft in human. Recent cryo-electron microscopy (cryo-EM) structure of yeast PIC reveals the closed clamp with a distorted promoter DNA in the closed cleft, which suggests that the transition of the clamp from the open conformation to the closed conformation induces DNA distortions to prime DNA melting[10]. In the OC structures of both human and yeast, the cleft is closed with the closed clamp conformation. In contrast, the biophysical study of archaeal transcription complexes using a single-molecule fluorescence energy transfer (smFRET) assay suggested a closed cleft PIC and open cleft OC[11]. Together, the status of the cleft during the transcription initiation process for archaeal RNAP and Pol II are not established and the dynamic nature of the PIC structure[7,10] presents a challenge to answering this question.

TFEα is a conserved GTF in archaeal-eukaryal transcription systems[12,13] and is an essential factor in vivo[14]. It consists of an N-terminal extended winged-helix domain (eWHD) and a C-terminal zinc ribbon domain (ZRD), and the X-ray crystal structure of the eWHD was determined from *Sulfolobus sulfataricus* (*Sso*)[15]. TFEα enhances transcription efficiency of RNAP in vitro, particularly for weak promoters under sub-optimal conditions[16–18], facilitates the formation of PIC and promoter DNA melting to form OC[17], and also stabilizes OC[19]. In relation to the clamp conformation, the smFRET study suggested that TFEα allosterically aids in opening the clamp during OC formation[11]. However, the molecular mechanisms of the functions of TEFα remain elusive due to lack of information about its structural conformation in the transcription complexes.

TFIIE is a heterodimer complex of TFIIEα/β in human and Tfa1/2 in yeast. The N-terminal region of TFIIEα or Tfa1 is essential and corresponds to archaeal TFEα[20], while TFIIEβ or Tfa2 is conserved only in eukaryotes and is functionally redundant[6,13]. The interactions between Pol II and TFIIE in the PIC have been studied from the cryo-EM structures in human and yeast[7–9] but the dynamic and transient nature of the PIC limits the resolution of the structures. Yeast PIC structures reveal the closed clamp, and there is no direct interaction between TFIIE and the clamp[7,9], whereas the human PIC structure is in the open clamp, enabling TFIIE interaction with the clamp[8]. Human PIC

structure suggests that TFIIEβ interacts with the clamp[8]. Additionally, biochemical probes and crosslinking suggest that Tfa1 interacts with the clamp in PIC[21,22]. Thus, the molecular details of interactions between TFIIE and Pol II in the PIC remain unresolved and the molecular mechanism of how TFIIE is involved in the clamp conformation change is unknown.

In this study, we determined the X-ray structure of TFEα and the cryo-EM structures of three forms of archaeal RNAP, including the apo-form, RNAP–TFEα binary complex, and RNAP–TFEα–DNA ternary complex. The structures reveal that direct binding of TFEα can shift the clamp conformation from a closed conformation to an open conformation. Our results show dynamic structural interactions among TFEα, RNAP, and promoter DNA, which provides structural information for the TFEα functions and transcription mechanism.

## Results

**X-ray crystal structure of TFEα.** To determine the structure of TFEα, we first tried, unsuccessfully, to crystalize full-length TFEα from euryarchaea *Thermococcus kodakarensis* (*Tko*) and *Pyrococcus furiosus* (*Pfu*). After screening derivatives of TFEα, we found that the 13 kDa fragment *Pfu* TFEα (TFEαΔZRD, 1–110 amino acid residues) was crystallized. The structure was solved by a single wavelength anomalous dispersion (SAD) at 3.2 Å and the resolution of the structure was improved to 2.6 Å with a native crystal (Supplementary Table 1). The TFEαΔZRD structure consists of the N-terminal eWHD (residues 5–86) and a long α-helix (α5) that connects the eWHD and the ZRD (Fig. 1a, b). The eWHD structure is similar with the X-ray crystal structures of the eWHD of *Sso* TFEα[15] and the eWHD of human TFIIEα[23].

We find previously unrecognized features of the TFEα structure from the TFEαΔZRD structure. There were eight molecules in the asymmetric unit of the SAD structure and two molecules in the asymmetric unit of the native structure. Superposition of the globular part of the eWHD from the 10 crystallographically independent structures (average root-mean-square deviation in α-carbon positions are 0.369–0.589 Å over residues 8–62) reveal flexibilities in the wing and the α5 linker (Fig. 1c). The conformation of the wing can change about 24° (~13 Å), and the α5 can be bent at ~21° (~12 Å) for the eWHD (Fig. 1d). These results show a potential structural adaptability of TFEα during the transcription initiation process.

**Cryo-EM structures of apo-RNAP and the binary complex of RNAP–TFEα.** To investigate molecular details of the interactions among RNAP, TFEα, and promoter DNA during transcription initiation, we determined the cryo-EM structures of *Tko* RNAP in three distinct states in the transcription cycle (Fig. 2a, b). The three forms were the apo-form, the TFEα bound form (RNAP–TFEα binary complex), and the TFEα and DNA bound form (RNAP–TFEα–DNA ternary complex) at nominal resolutions of 3.9 Å, 4.0 Å, and 3.8 Å, respectively (Supplementary Table 2 and Supplementary Figs. 1–3).

We previously solved the X-ray crystal structure of *Tko* RNAP[2]. To resolve accurate structural changes of *Tko* RNAP in solution through interactions with TFEα and promoter DNA, we determined the cryo-EM structure of *Tko* apo-RNAP. Three-dimensional classification of the apo-RNAP dataset showed that 82% particles formed complete apo-RNAP structures whereas 18% particles formed a stalkless RNAP structure (Supplementary Fig. 1c). The class of apo-RNAP comprising 56%, which we refer to as a major class of apo-RNAP, was further refined and the resulting map showed a well-resolved core and structural periphery. We modeled the apo-RNAP structure into the cryo-EM map (Supplementary Fig. 1f) and compared it with the

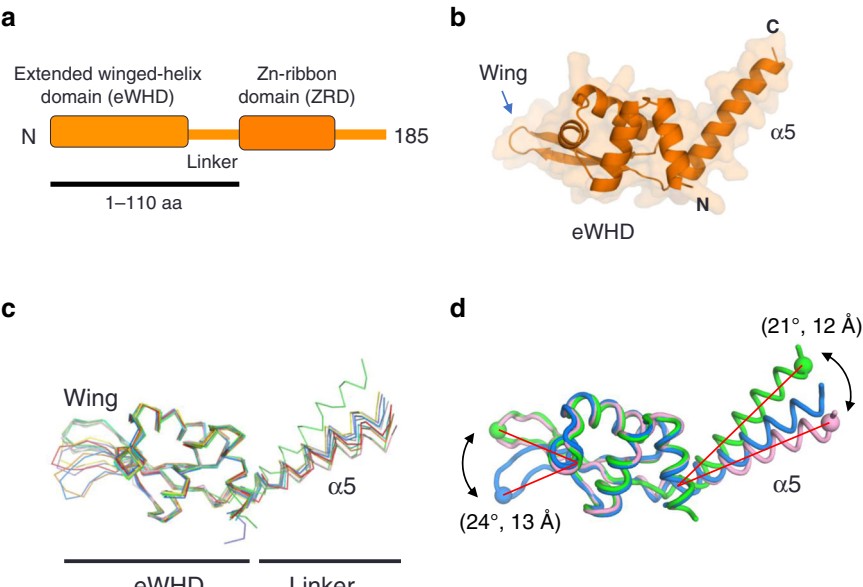

**Fig. 1 X-ray crystal structure of *Pfu* TFEαΔZRD. a** Domain organization of the TFEα. The amino acid region used for crystallization is underlined. **b** A ribbon model of TFEαΔZRD is shown with transparent surface. **c** Ten crystallographically independent TFEαΔZRD structures are shown as α-carbon backbone. The structures are superimposed via eWHD with wing excepted, showing flexibility in the orientations of the wing and linker. **d** Flexibilities of the wing and linker of TFEα. Three representative conformations of crystallographically independent TFEαΔZRD show swing motions of the wing and linker of TFEα.

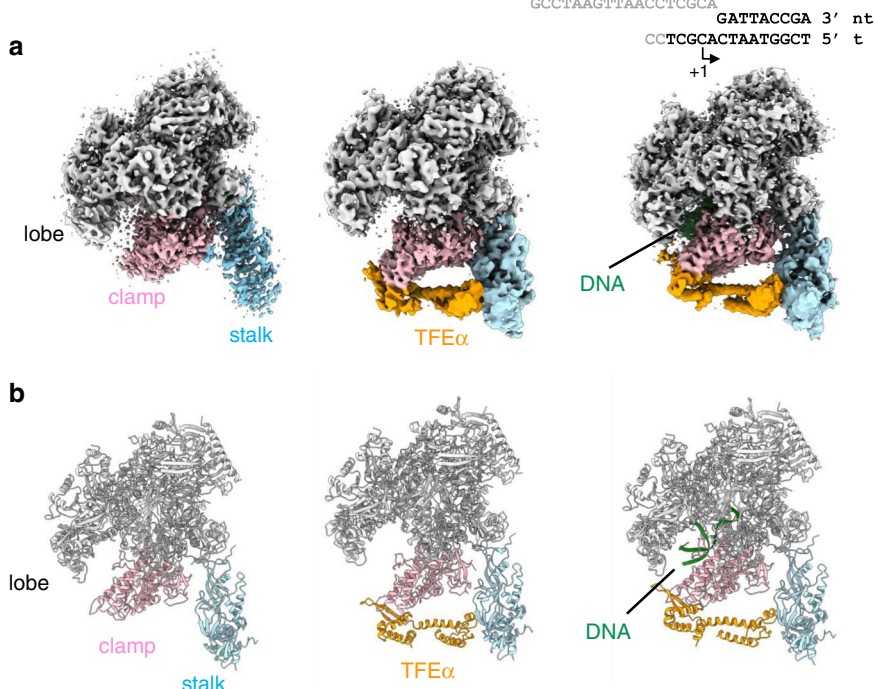

**Fig. 2 Cryo-EM structures of three different forms of the *Tko* RNAP. a** Cryo-EM density maps of the apo-form (left), RNAP–TFEα binary complex (center), and RNAP–TFEα–DNA ternary complex (right). Domains of RNAP and DNA are indicated and variably colored. The DNA used for preparing the ternary complex is shown (right). DNA bases shown in gray are disordered in the structure. Note that the cryo-EM maps of RNAP–TFEα binary complex and RNAP–TFEα–DNA ternary complex are the composite maps produced from globally refined maps and focused refinements on the region containing TFEα, stalk and cleft domains. **b** Ribbon models of the apo-form (left), binary complex (center) and ternary complex (right) in the same color and orientation as in (**a**).

previous X-ray crystal structures of archaeal RNAP[1,2]. We discovered that the cryo-EM structure of apo-RNAP adopts a closed clamp conformation (Supplementary Fig. 4). Another 3D class covering 22%, referred as a minor class, was also refined and

the resulting structure was similar with the major class structure with 3.3° shift in clamp conformation to the open direction compared to the major class structure, indicating that there is a slight mobility in the clamp of apo-RNAP. Consistent with this,

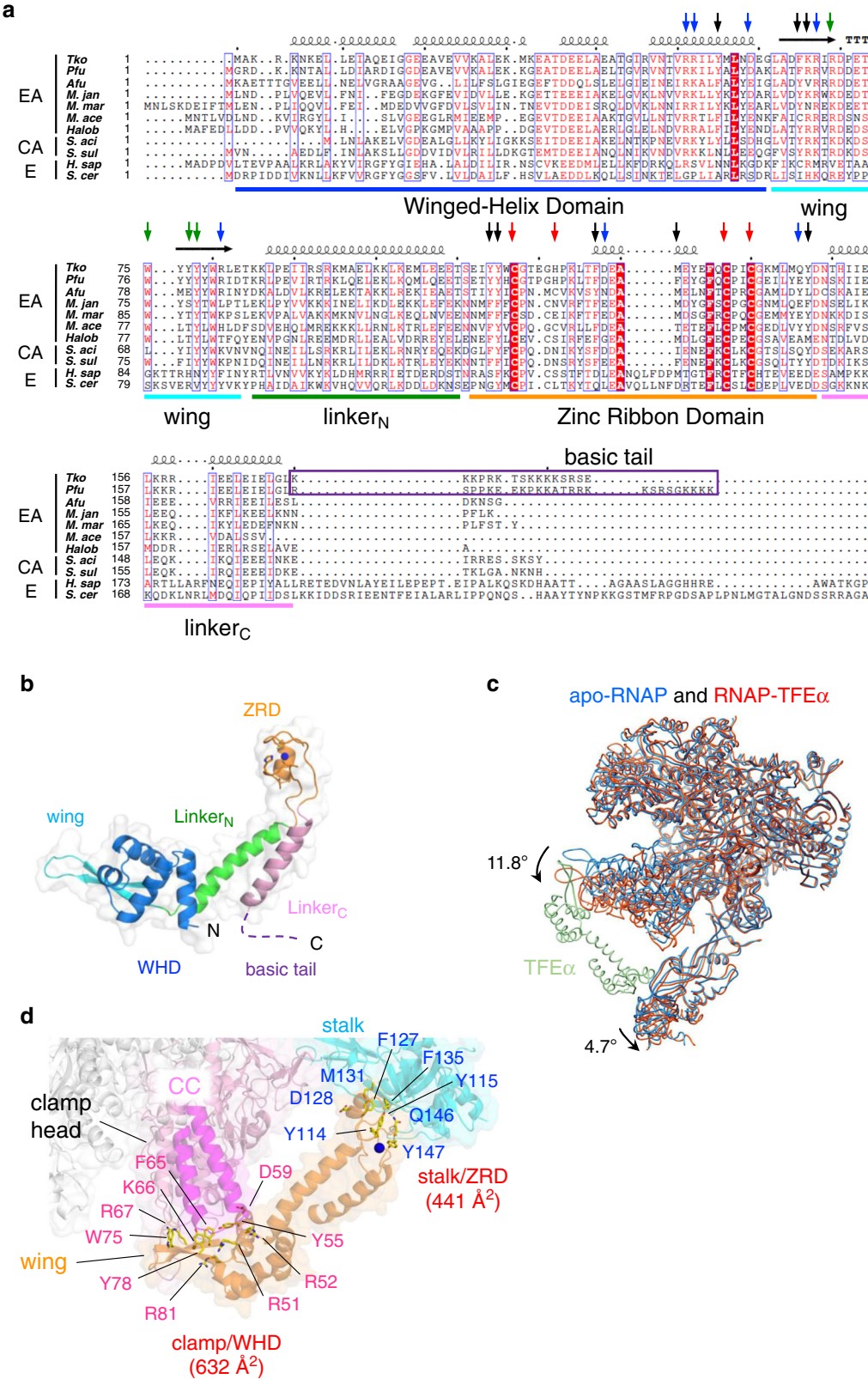

the results of the smFRET experiment suggested that free apo-RNAP in solution is predominantly in a closed clamp conformational state[11].

To visualize the effect of TFEα binding on RNAP and molecular details of the RNAP and TFEα interaction, we determined the cryo-EM structure of the RNAP–TFEα binary complex. The amino acid sequences of TFEα from *Pfu*, and *Tko* are 71% identical (Fig. 3a), which allowed us to use the X-ray structure of TFEαΔZRD determined in this study (Fig. 1b) as a guide to build the complete model of TFEα with the eWHD, ZRD, and two α helix bundles linking these domains (Figs. 2b and 3b). The overall shape of *Tko* TFEα structure resembles the

**Fig. 3 RNAP and TFEα interaction. a** Amino acid sequence alignment of euarchaeal (EA) and crearchaeal (CA) TFEα and eukaryotic (E) TFIIEα. The amino acid residues corresponding to the domains and motifs are indicated by bars and are the same color as in (A). Absolutely conserved residues are shown as white letters with red background, and highly conserved resides are indicated by red letters. Secondary structures of *Tko* TFEα are shown (α helix, coil; β strand, arrow). Amino acid residues involved in the RNAP interaction are indicated by arrows of the same color as in (B). *Tko*, Thermococcus kodakarensis; *Pfu*, Pyrococcus furiosus; *Afu*, Archaeoglobus fulgidus; *M.jan*, Methanocaldococcus jannaschii; *M.mar*, Methanococcus maripaludis; *M.ace*, Methanosarcina acetivorans; *Halob*, Halobacterium; *S.aci*, Sulfolobus acidocaldarius; *S. sol*, Sulfolobus solfataricus; *H.sap*, Homo sapiens; *S.cer*, Saccharomyces cerevisiae. **b** TFEα structures in the RNAP–TFEα binary complex are depicted as a ribbon model and transparent surface. Domains and motifs are indicated and colored as in (**a**). Disordered C-terminal tail is shown as a dashed line. **c** The structures of the apo-RNAP (blue) and the RNAP–TFEα binary (red) complex are superimposed. TFEα in the binary complex is in soft green and conformational changes of the clamp and stalk by TFEα binding are indicated. **d** Close-up view of the TFEα and RNAP interaction. The ribbon models of RNAP (clamp, pink; coiled coil (CC), magenta; stalk, cyan) and TFEα (orange) are shown with transparent surfaces. Amino acid residues involved in the RNAP and TFEα interaction are depicted as sticks and labeled (magenta and dark blue for residues interacting with the clamp and stalk domains, respectively). The contact surface areas between the eWHD and clamp and between the ZRD and stalk are indicated.

TFIIEα within the human PIC structure[8]. The distinction between the two is that TFEα in the RNAP–TFEα binary complex interacts directly with the tip of the clamp whereas there is no direct interaction between TFIIEα and the tip of clamp in the human PIC (Supplementary Fig. 5)[8]. For coordinating Zn atoms in the ZRD, *Tko* TFEα uses three Cys residues (C117, C137, and C140) together with a His residue (H122) instead of using four Cys residues in case of the eukaryotic TFIIEα (Supplementary Fig. 5a). Archaeal TFEα contains a positively-charged tail (16 residues) at its C-terminus but its density is not traceable (Fig. 3a, b).

To reveal the precise effect of TFEα binding on RNAP, we compared the cryo-EM structures of the apo-RNAP and RNAP–TFEα binary complex. In the latter, the clamp swings away from the DNA binding cleft at 11.8° relative to the apo-form RNAP, and this movement is coupled with the stalk swinging (4.7°) away from the clamp (Fig. 3c and Supplementary Movie 1). The structure shows that TFEα connects two mobile modules of RNAP, the clamp and stalk. It demonstrates that direct binding of TFEα on RNAP shifts the clamp and the stalk conformation from a closed state to an open state.

The RNAP–TFEα binary complex structure shows the molecular details of the interplays between RNAP and TFEα to open the DNA binding cleft of RNAP. Two α helixes between the eWHD and ZRD form an α-helix bundle linker, placing the ZRD over 50 Å away from the eWHD and allowing the eWHD and ZRD of TFEα to locate adjacent to the clamp and stalk domains of RNAP, respectively (Fig. 3d). The hydrophobic patch of the eWHD (residues Y55, F65, W75, and Y78) faces the tip of the coiled-coil (CC) of the clamp domain and the wing fits into the cavity between the CC and clamp head domain. Several hydrophilic residues of TFEα (R51, R52, D59, K66, R67, and R81) are also involved in the interaction with the RNAP clamp domain. The ZRD establishes its interaction with RpoE in the stalk domain of RNAP via hydrophobic surfaces (residues Y114, Y115, F127, M131, F135, and Y147). The D128 amino acid of TFEα also forms a salt bridge with K152 of RpoE. Consistent with the structure, deletion of the coiled-coil tip, amino acid substitutions at hydrophobic patch of RpoE (Y95E/F98E/L107E), or deletion of a short loop of RpoE (ΔS150-R157) disrupts TFEα binding to RNAP[16,24]. The contact surface areas between the eWHD and clamp and between the ZRD and stalk are 632 and 441 Å$^2$, respectively. Amino acid residues involved in the TFEα-RNAP interaction are conserved in archaeal TFEα and only a few are conserved in eukaryotic TFIIEα (Fig. 3a). This explains the much lower interaction affinity in eukaryotic Pol II counterparts.

**The structure of RNAP–TFEα–DNA ternary complex: DNA loading to the RNAP cleft closes the clamp.** To examine the conformation of archaeal RNAP with TFEα in the DNA bound

form, and how TFEα stabilizes the RNAP and DNA complex, we determined the cryo-EM structure of RNAP–TFEα–DNA ternary complex (Supplementary Table 2). We used a synthetic scaffold DNA in the form of a downstream fork junction promoter DNA containing the downstream double-stranded DNA and the partial transcription bubble (Fig. 2a), which functionally mimics open promoter DNA[25]. Strong densities are observed in the ternary complex for double-stranded DNA within the DNA binding cleft and 5 bases of single-stranded template DNA (from −3 to +2) positioned near the active site of RNAP. However, density for the non-template DNA in the bubble is not traceable (Fig. 4a). The DNA position within the RNAP DNA binding cleft is the same as that of the cryo-EM structures of the yeast and human OC[7,8].

To identify the structural changes of RNAP and TFEα upon the binding of the transcription bubble-mimicking promoter DNA, we compared the cryo-EM structures of the RNAP–TFEα binary complex and RNAP–TFEα–DNA ternary complex and found that DNA binding to the DNA binding cleft of RNAP changes the conformations of the clamp and stalk back to the closed state (Fig. 4b). Although the clamp and stalk swing 17° (14 Å) and 7.3° (3.7 Å), respectively, TFEα still maintains the same interactions with these domains, as observed in the RNAP–TFEα binary complex. Relative orientations of the eWHD and ZRD in TFEα change in the binary and ternary complexes (Fig. 4c). The structural flexibility of TFEα, including the α5 linker (Fig. 1c, d), allows it to bind the domains and function dynamically in both open and closed states.

Closing of the clamp establishes a new interaction between the TFEα wing and the lobe domain of RpoB, including the interaction of R69 side chain (TFEα) with Q285 side chain and G283 main chain at the lobe loop (Fig. 4d). This interaction encloses the DNA binding main cleft and provides a structural basis for TFEα stabilization of the OC, which is reminiscent of the mechanism for Spt4/5 stabilizing the elongation complex (Supplementary Fig. 6)[26].

**Models of the PIC and OC.** We modeled the archaeal PIC and OC (Fig. 5 and Supplementary Movie 2) using the cryo-EM structures of the RNAP–TFEα and RNAP–TFEα–DNA complexes determined in this study. For modeling DNA, TBP, and TFB, we used the cryo-EM structures of the yeast PIC containing a straight B-form DNA (PDB: 6GYK)[10] and human OC (PDB: 5IYB) containing both the template and non-template DNA strands in the transcription bubble[8] as templates. Cryo-EM structures of the *Tko* RNAP–TFEα (open clamp form) and RNAP–TFEα–DNA complexes (closed clamp form) were aligned with Pol II in the PIC and OC, respectively, using the catalytic core of RNAP containing the double-phi-β-barrel (DPBB) domains of the largest (RpoA1/Rpb1) and the second largest (RpoB/Rpb2) subunits.

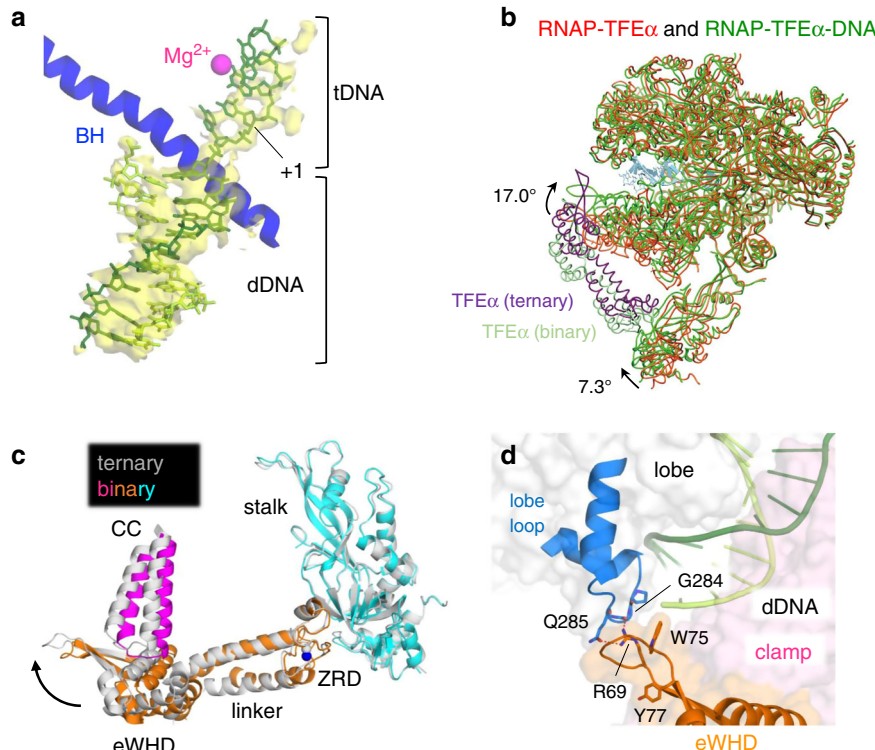

**Fig. 4 DNA interaction with RNAP in the presence of TFEα. a** Close-up view of the DNA binding channel of the RNAP–TFEα–DNA complex is shown with the cryo-EM density map (yellow and transparent) of DNA. DNA is shown as a stick model (template DNA, dark green; non-template DNA, light green). Template DNA strand (tDNA), downstream double-stranded DNA (dDNA) and transcription start site (+1) are indicated. $Mg^{2+}$ bound at the active site of RNAP and bridge helix (BH) are depicted. **b** The structures of the RNAP–TFEα binary (red) and the RNAP–TFEα–DNA ternary (green) complexes are superimposed. The conformational changes of the clamp and stalk by DNA positioning in the ternary complex are indicated. **c** Conformational change of TFEα upon RNAP binding DNA. The ZRD of TFEα was used as a reference to superimpose the RNAP and TFEα in the binary (color) and ternary complexes (light gray), revealing movement of the eWHD of TFEα together with the RNAP clamp indicated by an arrow. **d** Close-up view of the RNAP lobe and TFEα wing interaction. Amino acid residues involved in the RNAP lobe and TFEα wing interaction are depicted as stick models and indicated. Surface exposed aromatic residues (W75 and Y77) of TFEα involved in the DNA interaction are also indicated.

In the PIC model, the open clamp state of RNAP can accommodate double-stranded DNA between the TFEα wing and RpoB lobe loop (Fig. 5a), whereas clamp closure results in a steric clash between the TFEα wing and DNA. The lobe loop positions right below a major groove of DNA around −3T/−2T position, whereas the TFEα wing does not establish any contact with DNA. Thus, the opening state facilitates the subsequent positioning of the double-stranded downstream DNA into the DNA binding cleft, which explains the transcription enhancing activity of TFEα.

In the OC model, the RNAP clamp can be closed without steric clash with single-stranded DNA strands in the transcription bubble (Fig. 5b). The TFEα wing is located adjacent to the non-template strand around the upstream edge of the transcription bubble, which is similar to the eukaryotic OC structures[7,8]. We observe that β2 of the wing is on the side of the non-template DNA and the aromatic side chain of W75 is positioned to establish the stacking interaction with the base of the non-template DNA at −9 position where the DNA bubble starts. This is reminiscent of the interaction between the bacterial RNAP σ factor region 2.3 and the non-template DNA in the open promoter complex, RPo[27]. This suggests a molecular mechanism of how TFEα stabilizes the transcription bubble during transcription initiation by the interaction with the upstream DNA of OC[19].

**Crosslinking experiments in the archaeal PIC and OC.** To investigate how TFEα interacts with promoter DNA and the non-template DNA strand observed in the OC model, we used a highly specific UV-inducible crosslinking system based on the site-specific incorporation of unnatural amino acid para-benzoyl-phenylalanine (Bpa)[28] (Fig. 6). We prepared a series of Bpa substituted *Pfu* TFEα variants, including Bpa substitutions at I17 and N49 in the globular part of eWHD and Y56, F66, R70, W76, Y78, and Y79 in the wing (Fig. 6a). The Bpa reacts preferentially with unreactive C–H bonds when exposed to long wave UV light, with a reactive spherical radius of 3.1 Å[29], and these covalent cross-links can provide very specific proximity information for interacting macromolecular surfaces.

First, to establish whether the TFEα Bpa substitutions affected transcription activation by TFEα, we examined each TFEα variant for *gdh* promoter-dependent transcription in transcription complexes containing TBP and TFB2 (Fig. 6c and Supplementary Fig. 7). TFB2 was used since TFEα activation is greater with TFB2 than with TFB1[30]. A modest activation was observed for wild type TFEα and all Bpa variants except for Y79Bpa, which lacked any activation, possibly because of interference by the bulkier Bpa R-group. This indicated that the transcription activation by TFEα was not affected by Bpa substitutions in most variants.

Next, to monitor the site-specific contact between TFEα and DNA, we prepared three sets of *gdh* promoter DNA containing radiolabel at −9/−8/−7 positions of non-template DNA (−9NT), at −3 position of non-template DNA (−3NT), and at −4 position of template DNA (−4T) (Fig. 6b). The −9NT and −3NT/−4T promoter positions are located at the upstream edge and at the middle of the DNA bubble in the open complex of the eukaryotic Pol II transcription system[7,8]. Previous DNA to protein cross-links

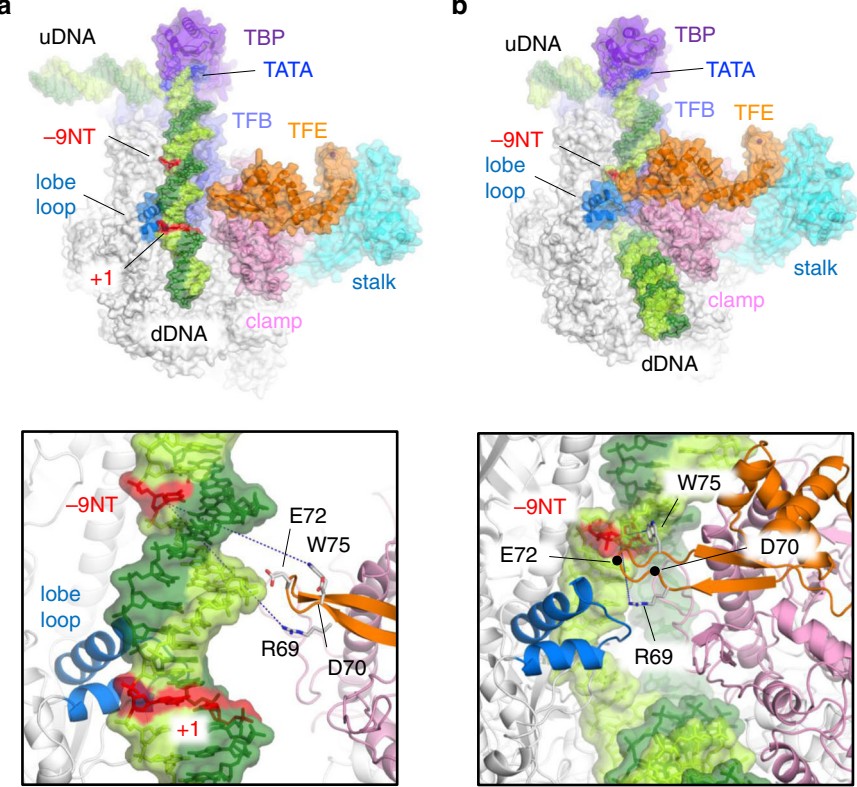

**Fig. 5 Structural models of archaeal PIC and OC models.** Overall views of the PIC (**a**) and the OC (**b**). RNAP (white with pink, cyan, and light blue), TBP (purple), TFB (light purple) and TFEα (orange) are depicted as ribbon models with transparent surface. DNA is depicted as stick model with transparent surface (template DNA, dark green; non-template DNA, light green; TATA box, blue). Positions of −9 base of non-template DNA (−9NT) and transcription start site (+1) are indicated. Close-up views of DNA from −9 to +1 positions in the PIC and RPo are shown in boxes. Amino acid residues involved in the DNA interaction are depicted as stick models and labeled. TFB is removed to clarify the view.

indicated that TFEα in preinitiation complexes cross-links strongly to −11 and −9 of the NT strand, but not to the T strand[19,31]. We observed crosslinking signals for the DNA radiolabelled at −9NT position with TFEα variants with Bpa at R70, W76, Y78, and Y79 (Fig. 6d and Supplementary Fig. 8). The strongest signal was observed in W76Bpa, located at the tip of wing, indicating that W76 is positioned proximally to the upstream edge of transcription bubble. This result is consistent with the observation in the OC model (Fig. 5b) that the corresponding residue W75 is located to interact with the non-template DNA at the upstream edge of transcription bubble. In the OC model, when the clamp is open, the distance between −9NT base and W75 of TFEα is 21 Å, which is too far for crosslinking, whereas the distance becomes 10 Å when the clamp is closed. Thus, the crosslinking result also indicates that the RNAP clamp is closed in the OC.

## Discussion

Here, we report the cryo-EM structures of apo-RNAP, RNAP–TFEα binary complex, and RNAP–TFEα–promoter DNA ternary complex that visualize dynamic structural changes of TFEα and archaeal RNAP, especially in the clamp and stalk, during transcription initiation (Supplementary Movies 1). Stable binding of TFEα to *Tko* RNAP without other GTFs and promoter DNA[32] allowed us to demonstrate that the direct binding of TFEα opens the DNA binding cleft of RNAP. We suggest a simplified transcription initiation model (Supplementary Movie 2) in archaea based on the model structures of PIC and OC in this study (Fig. 5). In this model, the clamp of apo-RNAP is closed conformation with a slight mobility. In the PIC, the binding of

TFEα on the clamp and stalk shifts the conformations of the mobile domains of RNAP to an open state (Fig. 5a) and then, the double-stranded downstream promoter DNA is placed into the DNA binding cleft. The positioning of the downstream DNA into the DNA binding cleft induces the clamp closing by strong ionic interactions between positively-charged residues covering the inside of the DNA binding cleft and a negative-charged DNA backbone. During the clamp closure, DNA distortion may be induced as observed from the yeast Pol II system[10], and we suggest that the wing of TFEα may be inserted into the DNA minor groove around −6 position where the transcription bubble is formed (Supplementary Movie 2). This also explains how TFEα facilitates the unwinding of the DNA. In the OC, the clamp is in a closed state and the wing of TFEα is located to stabilize OC structure by its interactions with the opposite region of the cleft (Supplementary Fig. 6) and the non-template strand DNA at the upstream edge of the transcription bubble (Fig. 5b).

This model shows how the DNA binding cleft of stalk-containing RNAPs can be opened in PIC and provides the structural basis of TFEα functions to facilitate transcription initiation. A previous study using smFRET assay suggested TFEα helps clamp opening allosterically during OC formation[11]. However, our results show that the direct binding of TFEα opens the clamp conformation. In the cryo-EM structures of eukaryotic Pol II PIC, the interaction between Pol II and TFIIE was not resolved in relation with the conformation of the clamp and stalk because of the dynamic and transient nature of PIC structure as well as the complex composition[7–10] (Supplementary Fig. 9). Given that the essential N-terminal part of TFIIEα in humans (Tfa1 in yeast) corresponds to TFEα[20] and their structures are

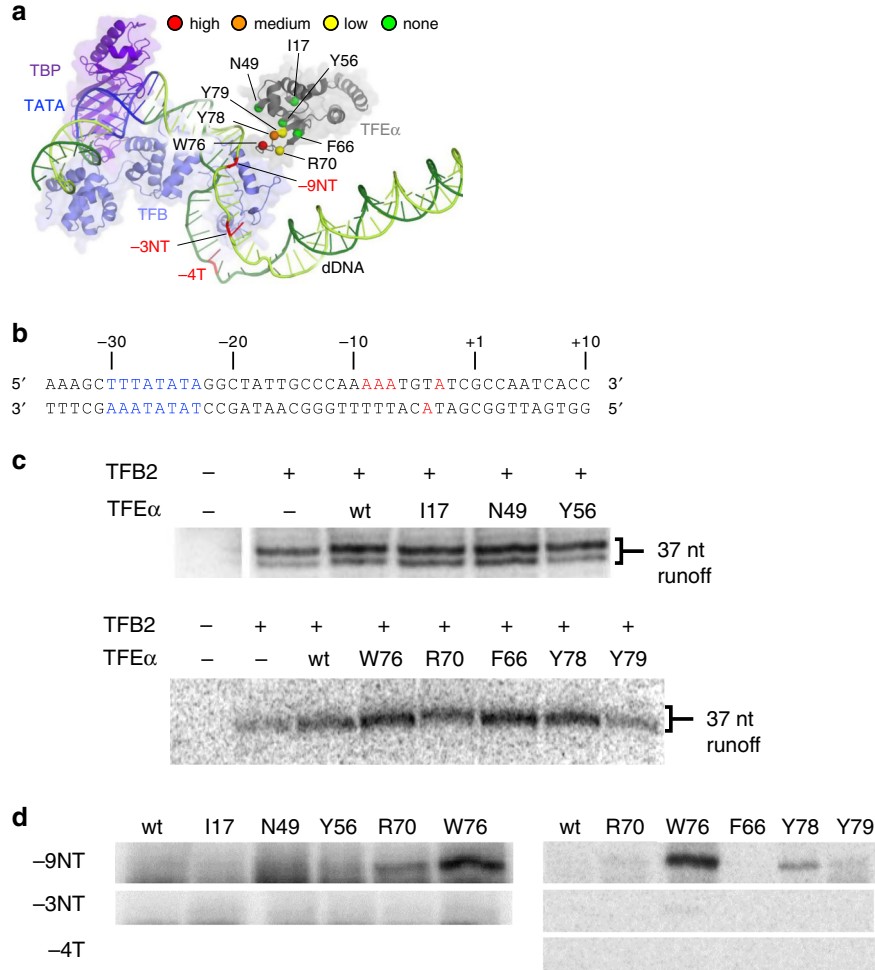

**Fig. 6 TFEα and DNA interaction investigated by photo-crosslinking. a** Structures of TBP, TFB, TFEα, and DNA in the OC model (Fig. 5b). Bpa substituted amino acids of TFEα (*Pfu* numbering) are indicted. Colors indicate cross-linking efficiency to DNA position at −9NT. Positions at −9NT, −3NT, and −4T are indicated. **b** DNA sequence of the *gdh* promoter used for photo-crosslinking. TATA box and radiolabelled positions are highlighted by blue and red, respectively. **c** Effects of Bpa substitution on transcription. Transcription of the *gdh* promoter was done in the absence (--) or presence of TFB2 and wild type and Bpa-substituted TFEα variants. Two sets of experiments are shown. In the upper set, run-off transcription gave rise to a primary transcript and a slightly shorter transcript, likely due to heterogeneity in the downstream end of the *gdh* promoter PCR product used. Full scan of the figures is in Supplementary Fig. 7. Each TFEα variant was tested at least two times, with representative results shown. **d** Crosslinking of TFEα Bpa variants to the *gdh* promoter. Preinitiation complexes were formed using *gdh* promoter variants radiolabelled on the NT strand (−9,−8,−7 or −3) or the T strand (−4), and cross-linking triggered by UV light. Cross-linked DNA was processed by nuclease and TFEα variants that retained radiolabel (indicating proximity between Bpa and radiolabel) were identified by SDS-PAGE. The region of the gels spanning ~25 to 30 kDA is shown. Full scan of the figures is in Supplementary Fig. 8. Each TFEα variant was tested with each radiolabeled *gdh* promoter variant at least three times, with representative results shown.

conserved (Supplementary Fig. 5), our results suggest that the clamp opening activity of TFEα in the PIC by the direct interaction with the tip of the clamp is conserved in TFIIEα (Tfa1 in yeast) as well. This is consistent with other studies[8,33] where the intermediate PIC structure having an open clamp conformation has been observed in humans, and has been reported as a requirement for the priming of DNA melting[10]. Additionally, our results are concurrent with previous findings that the cross-linking assay of yeast PIC showing eWHD of Tfa1 is located close to the clamp in the PIC[22].

Our results also suggest how the binding of TFEα to RNAP can shift the conformation of the clamp and stalk from a closed state to an open state. The cryo-EM reconstruction of the RNAP–TFEα binary complex in the high threshold of the map shows that the density between the ZRD and the stalk is rigid, whereas the density of the eWHD and clamp is weak (Supplementary Fig. 10), which suggests a modest structural heterogeneity in the eWHD and clamp. The asymmetric interaction of TFEα on the stalk and

clamp shows the mode of TFEα binding on RNAP and suggests a model of how TFEα binding on the clamp and stalk domains shifts their conformations from a closed state to an open state. In this model, the interaction between the ZRD and the stalk tethers TFEα on the RNAP. Subsequently, the eWHD of TFEα interacts with the coiled-coil tip of the clamp, which physically connects the center region of the stalk and the tip of the clamp. The bridging of the two domains of RNAP by TFEα mechanically allows the conformational change of the clamp from a closed state to an open state via a leverage effect. There is a slight structural heterogeneity in the clamp and eWHD due to the flexibility of the domains. The conformational change in the clamp produces a coupled conformation change in the stalk, as observed previously in other RNAP structures[2,34].

In eukaryotic Pol II, a clash of synthesized RNA with TFIIB causes promoter escape[35] and the subsequent initiation-elongation transition alters the path of upstream DNA without significant conformational change in Pol II[7,35,36].

The conformation of RNAP in the RNAP–TFEα–DNA ternary complex is in the closed clamp state (Fig. 4b) as archaeal RNAP in the RNAP–Spt4/5 complex structure[26] as well as Pol II in the OC structure[7–9] and the transcription elongation complex (EC) structure[36]. TFEα can bind to RNAP stably in the RNAP–TFEα–DNA ternary complex structure (Figs. 2 and 4), which supports the notion that TFEα is a part of early EC[19] and Spt4/5 displaces TFEα in the EC to enhance transcription processivity of the EC[16].

smFRET study in solution characterized the conformational positioning of the archaeal RNA polymerase clamp during the transcription cycle[11]. The structural information for the clamp conformation in this report is consistent with the smFRET work (Figs. 3c and 5a) but there is a difference after DNA melting and the OC formation (Figs. 4b and 5b). Recent reports showed that the OC formation is a multi-step process and clamp conformation changes are crucial event during DNA melting and the OC formation[37,38]. The structures of complete archaeal transcription complexes including GTFs such as TBP and TFB along with further biophysical studies would give refined models of transcription initiation mechanism in archaea.

The simpler bacterial system suggested detailed models for promoter DNA melting and subsequent arrangement of the transcription bubble[37–39]. Biophysical study assigned bacterial RNAP clamp conformation at each steps in transcription initiation and elongation[39] and real-time fluorescent study monitored that a transient clamp closure is required to nucleate DNA melting[37]. Universal structural features of RNAP within the DNA binding cleft play key roles in the late steps of promoter DNA melting[38] and clamp opening is required for the template DNA strand to enter into the active site of RNAP[37,38]. Our results support the universally conserved mechanisms of transcription initiation in all three domains of life by showing that direct binding of TFEα, a conserved GTF in archaeal–eukaryal RNAP systems, can open the clamp of the stalk-containing RNAPs. In *Thermus* and *Mycobacterium*, interestingly, an additional GTF CarD stabilizes the OC at the upstream edge of the transcription bubble[40,41] as TFEα does in archaea, which also suggests some domain- or lineage-specific transcription factors incorporated during evolution to carry out common functions.

## Methods

**TFEα cloning, expression, and purification**. The DNA fragment encoding the N-terminal 110 amino acids of the *Pfu* TFEα (TFEαΔZRD) was cloned into a pSUMO vector (LifeSensors) with BsaI/XhoI restriction sites (Supplementary Table 3). The plasmid was transformed into BL21(DE3) CodonPlus cells (Stratagene), and the transformed cells were grown in 2×YT media at 37 °C until $OD_{600}$ reached 0.6. Protein expression was induced by adding 0.2 mM isopropyl β-D-1-thiogalactopyranoside (IPTG) for 20 h at 20 °C. Harvested cells were suspended in buffer A (10 mM Tris-HCl (pH 8.0), 500 mM NaCl, 10 mM imidazole, and 5 mM β-mercaptoethanol) supplemented with 1 mM phenylmethylsulfonyl fluoride and lysed by sonication. After centrifugation, the supernatant was loaded onto a 5 ml Ni-NTA column (Qiagen) equilibrated with buffer A. After washing with 20 column volumes of buffer A containing 20 mM imidazole, proteins were eluted with buffer A containing 200 mM imidazole. The N-terminal (His)$_6$-SUMO tag was removed by the Ulp1 protease (LifeSensors) (1:200 of Ulp1:TFEαΔC) for 1 hour at 20 °C. After dialysis against buffer A without imidazole, the TFEαΔC was collected as the flow-through of a second Ni-NTA column. The protein was purified by heat treatment (70 °C for 30 min) and further purified by Superdex-200 size-exclusion column (GE Healthcare) equilibrated with buffer B (10 mM Tris-HCl (pH 8), 300 mM NaCl, 5% glycerol, and 5 mM dithiothreitol (DTT)).

Selenomethionyl (SeMet) *Pfu* TFEαΔZRD was prepared for the Single-wavelength anomalous diffraction (SAD) method. In addition to two original methionine residues at 1 and 106 positions, three methionines were introduced at residues Ile13, Ile54, and Phe66 positions using QuikChange site-directed mutagenesis (Stratagene). SeMet substituted protein was generated by growth in M9 minimal media supplemented with SeMet via the inhibition of methionine biosynthesis[42], and purified as the native protein.

**Crystallization and X-ray data collection**. For crystallization trials, native and SeMet substituted *Pfu* TFEαΔZRD were concentrated to 6 mg/ml. Crystals were grown within 2 days by hanging-drop vapor diffusion method at 21 °C with the crystallization solution containing 10% PEG 8000, 0.2 M $MgCl_2$, and 0.1 M Tris-HCl (pH 7). Crystals were cryo-protected by adding ethylene glycol to the reservoir solution to a final concentration of 20 % prior to plunge-freezing in liquid nitrogen. Diffraction data were collected at the A1 beamline of the Cornell University High Energy Synchrotron Source (CHESS, Ithaca, NY) at 100 K and data were processed by HKL2000[43]. Native crystals belonged to $P4_12_12$ and contained two molecules in an asymmetric unit, whereas SeMet labeled crystals belonged to $P2_12_12_1$ and contained eight molecules in an asymmetric unit.

**X-ray crystal structure determination and refinement**. With the anomalous signal from SeMet, 32 selenium sites in the asymmetric unit (4 sites in each of the 8 molecules in an asymmetric unit) were located and the experimental phase (figure of merit: 0.390) was calculated using Automated structure solution (AutoSol) in PHENIX[44]. Density modification by Automated model building (AutoBuild) in PHENIX yielded an excellent map and most of the model was built automatically. Manual model building was done using Coot[45] nm, and the structure was refined using PHENIX. The structure of native *Pfu* TFEαΔZRD was determined by the molecular replacement using Automated molecular replacement (Phaser-MR) in PHENIX followed by the structure refinement. Final coordinates and structure factors have been deposited to the Protein Data Bank (PDB) with the accession codes listed in the Supplementary Movie 1.

**Preparations of RNAP, RNAP–TFEα and RNAP–TFEα–DNA complexes**. *Tko* RNAP was purified from Δ*rpo4* strain KUWLFB[16] and then the 11-subunit apo-*Tko* RNAP was reconstituted in vitro by supplementing recombinant Rpo4 and Rpo7[2]. About 50 g of cells were suspended in 200 ml lysis buffer (10 mM Tris-HCl (pH 8.0), 500 mM KCl, 10% glycerol, 10 mM imidazole, 10 μM $ZnCl_2$, 5 mM 2-mercaptoethanol, 0.3 μM leupeptin, 1 μM pepstatin, 1.5 mM benzamidine hydrochloride and 0.5 mM phenylmethyl sulphonyl fluoride) and lysed by sonication. The supernatant was loaded to Ni-NTA affinity columns (Qiagen) equilibrated with the lysis buffer and washed with the same buffer containing 20 mM imidazole. Proteins were eluted with the lysis buffer containing 200 mM imidazole and precipitated by ammonium sulfate (final 80% saturation). The pellet was suspended in TGED buffer (20 mM Tris-HCl (pH 8.0), 10% glycerol, 0.5 μM EDTA and 5 mM DTT) and RNAP was further purified by binding and elution from HiTrap Q HP (GE Healthcare). To reconstitute RNAP containing all subunits, RNAP pools from HiTrap Q were mixed with recombinant Rpo4/Rpo7 and Rpo4 at a ratio of 1:4:1 (RNAP:Rpo4/Rpo7:Rpo4) for 1 h at 20 °C and were further purified by successive passage and elution from HiTrap Heparin and Superdex200 columns (GE healthcare). The gene encoding *Tko* TFEα was cloned into a pSUMO vector (LifeSensors) with BsmBI/BamHI restriction sites (Supplementary Table 3) and *Tko* TFEα was expressed in BL21(DE3) CodonPlus cells (Stratagene) using a pSUMO vector (LifeSensors) in 2×YT media at 37 °C. The temperature was shifted to 18 °C when $OD_{600}$ of culture reached 0.6 and protein expression was induced by 0.2 mM IPTG for 20 h. After harvesting cells, TFEα was purified as TFEαΔZRD.

The binary complex of RNAP and TFEα was formed by mixing 3 nmole RNAP with 5-fold molar excess of TFEα at room temperature for 1 h. The sample was applied to a Superose 6 10/300 GL column (GE Healthcare) equilibrated with 10 mM Tris-HCl (pH 8.0), 400 mM NaCl, 5 mM DTT, 1% glycerol, 0.1 mM EDTA and collected in 300 μl fractions. The fraction corresponding to the peak of the complex, in about 0.3 mg/ml, was used directly in the subsequent EM analysis.

For the preparation of RNAP–TFEα-promoter DNA ternary complex, DNA scaffold based on the *Tko glutamate dehydrogenase* (*gdh*) gene promoter (template strand: from −4 to +12, 5′-TCGGTAATCACGCTCC-3′, non-template strand: from −15 to +12, 5′-GCCTAAGTTAACCTCGCAGATTACCGA-3′) was used. Duplex DNA was generated by mixing equimolar single-stranded DNA oligonucleotides in $H_2O$ and heating to 95 °C for 5 min. The sample was cooled to 10 °C at a rate of 1 °C per min.

The ternary complex of RNAP, TFEα, and DNA was prepared by mixing 3 nmol binary complex of RNAP–TFEα with 3-fold molar excess of the annealed DNA at room temperature for 30 min. The ternary complex was purified with a Superose 6 10/300 GL column (GE Healthcare) equilibrated with 10 mM Tris-HCl (pH 8.0), 400 mM NaCl, 5 mM DTT, 1% glycerol, 0.1 mM EDTA. A fraction of the peak of the complex in about 0.3 mg/ml was used in the EM analysis.

**Cryo-electron microscopy data acquisition**. Samples of 4 μl of purified apo-RNAP, RNAP–TFEα, and RNAP–TFEα–DNA of ~0.3 mg/ml were loaded onto freshly glow-discharged holey carbon grids (Quantifoil R1.2/1.3 Cu200, Quantifoil Micro Tools GmbH, Germany) followed by plunge freezing using a Vitrobot Mark IV (Thermo Fisher Scientific Inc., USA) at 90% relative humidity and 4 °C. For the RNAP–TFEα–DNA sample, from which a strong orientation preference was observed, additional data acquisition was performed using the sample loaded onto a holey carbon grid coated with a graphene oxide film. Image data collection was performed using a Titan Krios transmission electron microscope (Thermo Fisher Scientific Inc., USA) operating at 300 kV acceleration voltage, and equipped with image spherical aberration (Cs) corrector (CEOS GmbH, Germany) and a Falcon II direct electron detector. Imaging was performed in nano-probe mode, spot size 3, 70 μm C2 aperture and 100 μm objective lens aperture. Cs corrector was tuned just

before data acquisition at normal magnification of ×125,000 in order to minimize 2-fold and 3-fold astigmatism, axial coma, and spherical aberration. Automated data collection was performed using EPU software (Thermo Fisher Scientific Inc., USA) at a nominal magnification of ×47,000, which corresponds to a pixel size of 1.4 Å at the specimen scale, and to a defocus range between −1.5 μm and −3.5 μm. Thirty frames per movie data were collected at a dose rate of ~35 e−/Å²/s over a 1.8 s exposure time. A total of 1386 movies were collected for apo-RNAP and 2698 movies for RNAP–TFEα, respectively. For RNAP–TFEα–DNA complex, 1,737 movies and 2,106 movies were collected from the holey grid with and without additional graphene oxide support film, respectively.

**Cryo-electron microscopy image processing**. MotionCor2[46] was used for beam-induced movement correction and dose-weighing from the movie data. The motion-corrected micrographs were subjected to contrast transfer function estimation using Gctf[47]. Relion 2.1[48] was employed for subsequent image processing procedures, unless otherwise stated. Typically, 10,000–20,000 particles were manually picked from micrographs with high underfocus to aid visibility of the particles. The particles were extracted into 176 × 176 pixel boxes followed by 2D class averaging. Class averages with pronounced structural features were used as a template for automated particle picking. The box size and particle-picking regime were commonly applied to all datasets.

For apo-RNAP, 514,007 particles were extracted from particle auto-picking. Two rounds of 2D class averaging were performed from which particles that belonged to class averages with poor structural features were eliminated after each run, leaving 350,067 particles in the dataset (Supplementary Fig. 1b). The X-ray structure of *Tko* RNAP (PDB: 4QIW) was converted to an EM map and low pass filtered to 60 Å, using the *e2pdb2mrc.py* routine in the EMAN2 software package[49]. The volume was used as a reference for the subsequent 3D reconstruction process, but due to the strong preferential orientation of particles, the initial attempt of 3D reconstruction resulted in a poorly resolved map. To alleviate the effect of the preferred orientation, particle images that belonged to strongly preferred orientations were discarded based on the 2D class averages. The resulting 247,663 particles were used to reconstruct a consensus map without any preceding 3D classification. Then, the particles were classified into four 3D classes without performing any image alignment. The 139,242 particles that belonged in the 3D class that exhibited the best structural detail were selected for subsequent per-particle motion correction and B-factor weighing via particle polishing (Supplementary Fig. 1c). Finally, the corrected particle images were refined and post-processed to yield a reconstruction at 3.91 Å resolution based on a 0.143 Fourier shell correlation (FSC) cut-off criterion (Supplementary Fig. 1e)[50].

For RNAP–TFEα, a total of 1,226,339 particles were auto-picked. Then 953,770 particles belonging to the best 2D class averages were selected after two rounds of class averaging (Supplementary Fig. 2b). Using low pass filtered X-ray structure of *Tko* RNAP as an initial model, the dataset was subjected to 3D classification. Out of six resultant 3D classes, two classes exhibited the best structural details and unambiguously discernible densities for TFEα and stalk domain, and 409,653 particles belonging to those classes were selected for subsequent processing. The particles were re-classified in 3D without alignment in order to sort out particle images of intact complex and with best structural details. From eight 3D classes, 252,508 particles that belonged to the best two 3D classes chosen for final 3D reconstruction (Supplementary Fig. 2c). Particle polishing, refinement and post-processing resulted in a 3D reconstruction at 4.04 Å resolution based on a 0.143 FSC cut-off criterion (Supplementary Fig. 2e). A binary mask that embraced the density corresponding to stalk and clamp domains of RNAP and TFEα was created from a preliminary atomic model. Focused refinement on this region was performed by continuing the last iteration of the 3D refinement run of the entire structure with the applied local mask. The overall resolution of the resulting map from focused refinement was 4.25 Å with a slight improvement in local resolution of the masked region (Supplementary Fig. 2e, f).

For RNAP–TFEα–DNA, a total of 1,388,666 particles were auto-picked from the combined micrograph datasets obtained from the vitrified grids with and without graphene oxide film. After two rounds of 2D class averaging, 911,020 particles were selected (Supplementary Fig. 3a) and subjected to 3D classification into four classes. A total of 669,840 particles belonging to the 3D classes with pronounced structural details and intact domains, TFEα and DNA, were used to produce a consensus 3D reconstruction. The dataset was further classified into eight 3D classes without image alignment. After careful inspection of the 3D classes, two classes with intact TFEα, stalk domain, DNA, and good structural details were chosen, from which 312,092 particles were extracted (Supplementary Fig. 3b). Among the particles included for the final reconstruction, 121,727 and 190,365 particles were from the micrographs obtained from grids with and without graphene oxide film, respectively. After particle polishing, refinement, and post-processing, a final map at 3.79 Å resolution was produced (Supplementary Fig. 3d). A binary mask that embraces TFEα, DNA as well as stalk and clamp domains was applied for focused refinement that continued from the last iteration of the 3D refinement of the entire complex. The overall resolution of the resulting map from focused refinement was 3.85 Å with a slight improvement in local resolution of the masked region (Supplementary Fig. 3d, e).

Local resolution estimation of resulting cryo-EM map was performed using MonoRes by submitting half maps for the final reconstruction to the Scipion web

tools[51]. FSC between the unfiltered half maps was calculated using Relion 2.1. In case of RNAP–TFEα and RNAP–TFEα–DNA, the composite maps were produced by combining the final maps with the maps resulting from focused refinements using *combine_focused_maps* tool in PHENIX. FSC between the composite map and the refined atomic model was calculated using *phenix.mtriage* routine in PHENIX (Supplementary Figs. 2f and 3e)[52]. EM map display and figure preparation was performed using USCF Chimera[53].

**Modeling of cryo-EM structures**. We built molecular models of the apo-RNAP, RNAP–TFEα binary complex, and RNAP–TFEα-promoter DNA ternary complex into the cryo-EM maps using an X-ray crystal structure of apo *Tko* RNAP (PDB: 4QIW)[2]. The X-ray crystal structure of *Pfu* TFEαΔZRD and that of human TFIIE (PDB: 5GPY)[23] were used as references. The *Tko* RNAP crystal structure was fitted to the map as a rigid body, and then subunits and the mobile clamp domain were fitted individually. We modeled and extended TFEα and DNA structures manually using Coot[45] and further real-space refined the structures against the B-factor sharpened cryo-EM maps from relion_postprocess with PHENIX suite[44] and validated using MolProbity[54] (Supplementary Table 2).

**In vitro transcription and cross-linking**. Mutant TFEα variants containing site-specific Bpa substitutions were made using the pSUP system[28,55]. Amber codons were created in an N-terminal 6×-His tagged TFEα gene at desired amino acid positions, by primer extension/DpnI mutagenesis. The mutated TFEα variants were expressed from inducible plasmids alongside pSUP-BpaRS-6TRN, in the presence of *p*-benzoyl phenylalanine. The TFEα variants were then purified by nickel chromatography. Transcription reactions were conducted using purified recombinant TBP, TFB2, and TFEα and native *Pfu* RNAP (containing a 6× His tag on subunit D, purified by nickel chromatography[56]). Proteins were assembled at the *gdh* promoter for 30 min at 65 °C, with RNAP at 10 nM, TBP at 60 nM, TFB2 at 120 nM, and TFEα at 200 nM[30]. Ribonucleotide triphosphates (500 μM GTP, CTP, and ATP, 10 μM [alpha-32P]UTP at ~40 Ci mmol−1) were added to start transcription, reactions were incubated for an additional 20 min at 65 °C, and then stopped by adding EDTA to 17 mM, along with a small amount of radiolabelled DNA marker (100 bp). Samples were phenol extracted and ethanol precipitated, and transcripts visualized on 14% polyacrylamide gels. See Supplementary Fig. 7 for full scans of the gels.

TFEα Bpa cross-linking in preinitiation complexes was tested using 1 nM site-specifically radiolabeled *gdh* promoter DNA, at 10 nM RNAP, 60 nM TBP, 120 nM TFB1, and 500 nM TFEα. Radiolabelled *gdh* promoter templates were made from biotinylated PCR products attached to paramagnetic streptavidin beads. The non-biotinylated strand was dissociated, and primers annealed and extended with radiolabelled dATP, washed, and fully extended with 500 μM unlabeled dNTPs. The fully extended products were released by restriction enzyme digestion, phenol extracted and ethanol precipitated, and then dissolved and quantified by scintillation counting. Reactions were assembled in 25 microliters overlaid with mineral oil, incubated for 30 min at 65 °C, and cross-links were formed by direct illumination with a handheld UV lamp (365 nm) at 2 cm distance for 60 min. DNA was digested with DNaseI and micrococcal nuclease (10 units of each), and samples were loaded on 4–20% gradient polyacrylamide gels[55]. See Supplementary Fig. 8 for full scans of the gels.

**Reporting summary**. Further information on experimental design is available in the Nature Research Reporting Summary linked to this paper.

## Data availability

The atomic structures of native and SeMet substituted *Pfu* TFEαΔZRD and their structure factors were deposited in the Protein Data Bank with the accession code 6PLN and 6XJF, respectively. The cryo-EM maps of *Tko* apo-RNAP, RNAP–TFEα complex, and RNAP–TFEα–DNA complex were deposited in the Electron Microscopy Data Bank with the accession codes EMD-9960, EMD-9961, and EMD-9962, respectively. Their corresponding models were deposited in the Protein Data Bank with the accession codes 6KF3, 6KF4, and 6KF9, respectively. Other data are available upon request. PDB files used for structure comparisons (Supplementary Figs. 5, 6, and 9: 5GPY, 3P8B, 5XON, 5IYA, 5FZ5, and 5FMF) are from the Protein Data Bank (https://www.rcsb.org).

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

## Acknowledgements

We thank the staff at the Jinju Bio-industry Foundation for supporting *Tko* cell fermentation and Macromolecular X-ray science at the Cornell High Energy Synchrotron Source (MacCHESS) for supporting the crystallographic data collection. The cryo-EM experiment was performed at the Cryo-EM facility in Electron Microscopy Research Center, Korea Basic Science Institute (KBSI) and supported by KBSI grant C030221 and C021820. We thank Carmen V. Amoah-Kusi for assistance with cross-linking and transcription assays. This work was supported by the Basic Science Research Program through the National Research Foundation of Korea funded by the Ministry of Education (NRF-2015R1D1A1A01059097 to S.-H.J.) and by the Ministry of Science, ICT & Future Planning (NRF-2016R1A5A1010764 and NRF-2017M3A9F6029755 to H.-S.C. and NRF-2020R1F1A1072050 to S.-H.J.) and NIH grants (R01 GM087350 and R35 GM131860 to K.S.M. and R15 GM083306 to M.S.B.).

## Author contributions

K.S.M., S.-H.J., M.S.B., and H.-S.C. conceived and designed the experiments. S.-H.J. and K.S.M. crystallized and solved TFEα crystal structure. S.-H.J., J.S.C., and H.K. prepared *Tko* RNAP and its complexes for cryo-EM. J.H. conducted cryo-EM sample preparation, data collection, and image processing. S.-H.J. and K.S.M. carried out cryo-EM model building. M.S.B. performed cross-linking and transcription assays. K.S.M.,

S.-H.J., H.-S.C., J.H., and M.S.B. wrote the paper. All authors discussed the results and commented on the paper.

## Competing interests

The authors declare no competing interests.
