## [Peer Review File · Nature Communications]

REVIEWER COMMENTS

Reviewer #1 (Remarks to the Author):

Jun, Hyun et al., present the first structures of archaeal RNAP initiation complexes. Structures of initiation complexes from eukaryotes and bacteria have been determined, but structural information regarding how archaeal initiation factors promote DNA loading by RNAP have been lacking. Using X-ray crystallography and cryo-EM, the authors have carefully determined how TFE α promotes DNA loading by RNAP. The combination of structures, modelling, and supporting biochemical experiments elucidate how TFE α modulates RNAP cleft opening in the presence and absence of DNA. The manuscript is well written, the movies are incredibly helpful for understanding the dynamics of the complex, and the figures are clear for the most part. This work will serve as a foundation to structurally elucidate the complete initiation process in archaea. There are a few concerns that the authors should address prior to publication. If these concerns are addressed, I strongly support publication in Nature Communications.

Major Concerns

-The authors crystallise the TFE α eWHD under native and SeMet conditions. The coordinates for the native condition have been deposited in the PDB, but the SeMet coordinates have not. Given that TFE α crystallises in two different space groups and this results in different observed structures within the unit cells, it is critical that both data sets are deposited.

-Figure 6 c-d: The transcription and cross linking assays likely support the major findings of this manuscript regarding cleft opening; however, the data presentation is confusing. In panel C, two different gel images are provided. It is worrying that in the left panel, there is a band doublet whereas in the right panel there is no doublet and the image contrast appears to be significantly different. The authors should comment on this and pick imaging conditions that appear similar.

Panel D is more concerning because the provided gel images have differing results (ie R70 and W76 mutants). The band intensity for R70 at -9,-8-7 is significantly different between the two gels. There also appears to be a doublet on the left panel and no doublet in the right gel. The authors should provide full images of the autoradiograms in the supplement and explain discrepancies between the gels. Statements about the reproducibility of the experiments should also be provided. It would also be helpful if the authors explained why they chose the -3, -4 labelling sites and not the -6 site that they suggest is used to open the DNA.

Minor Concerns

-line 36 (abstract)-Specify which archaea were used in this study

-line 55-57- The authors state that archaeal transcription machinery is similar to that found in eukaryotes. Many enzymes in archaea are more similar to their eukaryotic counterparts, so it seems reasonable to remove the clause at the beginning of the sentence.

-line 133-134- the statement about TFE α flexibility should not include RNAP because this only becomes apparent later in the text. The sentence should be modified or placed elsewhere in the manuscript.

-Provide an SDS-PAGE of purified components. It would also help if the authors provided a UV 260/280 trace and SDS-PAGE for the purification of the complexes used in cryo-EM.

-Consider performing CTF refinement in RELION 3.0/3.1 to improve cryo-EM resolutions/overall quality of maps. This is rather straightforward and usually improves map quality.

-Supplementary figures 1-3-for cryo-EM data, provide angular distribution plot for final maps

-Fig 2. State that displayed maps are composite maps in the figure legend

-Fig 3D. It would be helpful to distinguish interacting residues residing in the clamp head and stalk from TFE α . Instead of colouring the residues by hydrophobicity, it would help to colour them according to domain/subunit.

-Figure 4a- A closer view of the DNA and active site with representative densities would be more helpful than the current rendering. The remainder of RNAP can be left out of this figure.

-Figure 4c- provide an inset similar to panel b distinguishing which ribbon model is the binary versus ternary complex (grey ternary, coloured binary).
-Figure 4d- a more focused view of the interactions would help with understanding how the side chains form contacts. Only a portion of RNAP and TFE α need to be shown.
-Figure 5 boxes- right box- would be helpful if positions of D70 and E72 were indicated
-Figure 6a- Provide a colour panel for the relative cross linking efficiencies. A supplementary panel showing how these residues may interact with non-template/template DNA would be helpful.
-Methods- For transcription assay, authors use 200 nM TFE α and cross linking assays use 500 nM TFE α . Please provide an explanation for the concentration difference between the assays.
-Methods- Authors state that to obtain the ternary complex, graphene oxide grids were used to overcome preferential orientation issues. It is unclear if the data from holey carbon grids and the graphene oxide grids were combined. If so, it should be explicitly stated how many particles were used from each dataset.

Reviewer #2 (Remarks to the Author):

Review of Jun et al., 2020

General

This is an excellent paper by the Murakami lab that shed light on the structural biology of transcription in archaea, namely the partial X-ray structure of TFE alpha, and the cryoEM structures of the binary RNAP-TFE alpha and ternary RNAP-DNA-TFE alpha complexes. These structures have been sorely missing as they provide a hint for the structural basis of promoter melting and DNA template strand loading into the active site - this makes a substantial contribution to the field, and I think that the article will be of great interest to your readership.

Specifics

Page 4: 'Our results suggest molecular mechanisms of multiple functions of TFE α during transcription initiation.'

Due to the absence of the essential transcription initiation factors TBP and TFB in the structures determined by the authors, this is an overstatement. The solved RNAP-DNA-TFE alpha structure could just as well represent the RNAP in a conformation that represents the transcription elongation complex (TEC). In this respect its noteworthy that TFE alpha can be incorporated into TECs in vitro (J Biol Chem. 2007 Dec 7;282(49):35482-90. doi: 10.1074/jbc.M707371200).

Page 8: '[...] indicating that the clamp opening by TFE α is an obligatory step during the PIC formation.'

Transcription initiation is not strictly dependent on TFE, therefore the action 'by TFE' cannot be an obligatory step, it rather happens spontaneously but is enhanced or catalyzed by TFE.

Page 9: 'promoter-dependent transcription in transcription complexes containing TFB2 (Fig. 6c)' This assay has to be explained better. Why TFB2 and not TFB1. And why is TBP not mentioned here? See also below.

Page 10: 'We suggest a simplified transcription initiation model (Supplementary Movie 2) in archaea based on the model structures of PIC and OC in this study (Fig. 5)'

Without TBP and TFB, the terms PIC and OC can be misleading, it remains an assumption that these complexes represent initiation complexes

Page 12: 'The conformational change in the clamp produces a coupled conformation change in the stalk, as observed previously in other RNAP structures^{5,32}.' The article ends a bit abrupt, and the topic would benefit from additional elaboration. The authors should also include a final paragraph here to reflect on the structural differences between initiation and elongation complexes. I'm

happy to go along with that the RNAP-TFE-DNA complexes presented in this article are resembling initiation complexes, but without TBP/TFB they cannot strictly speaking be called just that. The results in the paper are excellent - a discussion of this topic would suffice to alleviate my concerns. Katsu published the first archaeal RNAP-Spt4/5 cryoEM structure in PNAS years ago.

- What are the similarities and differences between the two structural states of RNAP?
- What conformational changes in RNAP have to occur during promoter escape?
- How does the RNAP-TFE-DNA structure support the notion that TFE can associate with TECs?

The RNAP clamp mobility and positioning has been characterised in solution using smFRET, as cited earlier on in the manuscript. But differences remain, and it would be good if the authors could compare their new structural information on the background of the clamp opening and closure described in solution through the transcription cycle.

Finally, detailed models of promoter melting and DNA template strand loading have been proposed for the simpler bacterial RNAPs. The manuscript would benefit from a detailed comparison of these processes between the two systems. Bacteria do not use TFE-like factors but some, e.g. the Mtb RNAP, uses analogous factors such as CarD with interesting differences and commonalities.

Methods.

Page 17: 'In vitro transcription and cross-linking. Mutant TFE variants containing site-specific Bpa substitutions were made using the pSUP system as described^{28,46}. Transcription reactions were conducted using recombinant TBP, TFB2, and TFE and native RNAP, assembled at the *gdh* promoter as described previously, with RNAP at 10 mM, TBP at 60 nM, TFB2 at 120 nM, and TFE at 200 nM⁴⁷. TFE Bpa cross-linking in preinitiation complexes was tested using 1 nM site-specifically radiolabeled *gdh* promoter DNA, at 10 nM RNAP, 60 nM TBP, 120 nM TFB1, and 500 nM TFE α . Cross-links were formed and products analyzed as described previously⁴⁶.'

Do you really mean 10 millimolar RNAP? Probably 10 micromolar, or was it nanomolar? Why use TFB1 in the cross-linking assay, and not TFB2 as in the transcription assay? Not that this is a major problem, but please explain.

This is a nice piece work, well done Katsu & team!
Finn Werner

REVIEWER COMMENTS

Reviewer #1 (Remarks to the Author):

Jun, Hyun et al., present the first structures of archaeal RNAP initiation complexes. Structures of initiation complexes from eukaryotes and bacteria have been determined, but structural information regarding how archaeal initiation factors promote DNA loading by RNAP have been lacking. Using X-ray crystallography and cryo-EM, the authors have carefully determined how TFE α promotes DNA loading by RNAP. The combination of structures, modelling, and supporting biochemical experiments elucidate how TFE α modulates RNAP cleft opening in the presence and absence of DNA. The manuscript is well written, the movies are incredibly helpful for understanding the dynamics of the complex, and the figures are clear for the most part. This work will serve as a foundation to structurally elucidate the complete initiation process in archaea. There are a few concerns that the authors should address prior to publication. If these concerns are addressed, I strongly support publication in Nature Communications.

Major Concerns

-The authors crystallise the TFE α eWHD under native and SeMet conditions. The coordinates for the native condition have been deposited in the PDB, but the SeMet coordinates have not. Given that TFE α crystallises in two different space groups and this results in different observed structures within the unit cells, it is critical that both data sets are deposited.

As this reviewer requested, we deposited the structural coordinate and factor in the PDB, and stated its PDB ID (6XJF) in the Supplementary Table 1.

-Figure 6 c-d: The transcription and cross linking assays likely support the major findings of this manuscript regarding cleft opening; however, the data presentation is confusing. In panel C, two different gel images are provided. It is worrying that in the left panel, there is a band doublet whereas in the right panel there is no doublet and the image contrast appears to be significantly different. The authors should comment on this and pick imaging conditions that appear similar.

We appreciate the reviewer's comment. The doublet in the left hand experiment likely arises from heterogeneity in the downstream end of the PCR product used as a promoter. The lower signal observed in the right hand experiment (which was done using a different PCR product promoter template) was due to lower specific activity of radiolabel used in that experiment. However, this does not compromise the experiment since the comparison of transcript levels can be done internally with

no TFE α or wild type TFE α .

The legend to Figure 6 is revised as follows:

“Effects of Bpa substitution on transcription. Transcription of the *gdh* promoter was done in the absence (--) or presence of wild type and Bpa-substituted TFE α variants. Two sets of experiments are shown. In the left hand set, run-off transcription gave rise to a primary transcript and a slightly shorter transcript, likely due to heterogeneity in the downstream end of the *gdh* promoter PCR product used.”

Panel D is more concerning because the provided gel images have differing results (ie R70 and W76 mutants). The band intensity for R70 at -9,-8-7 is significantly different between the two gels. There also appears to be a doublet on the left panel and no doublet in the right gel. The authors should provide full images of the autoradiograms in the supplement and explain discrepancies between the gels. Statements about the reproducibility of the experiments should also be provided. It would also be helpful if the authors explained why they chose the -3, -4 labelling sites and not the -6 site that they suggest is used to open the DNA.

The cross-linking experiments shown were done using different sets of radiolabeled *gdh* promoter probes that had different activities, explaining the difference in band intensity between the two experiments. In the left panel, the radiolabel beneath the ~28 kDa TFE α band is from residual undigested probe DNA. The full images of the gels are shown in the new supplemental figure (Supplementary Fig. 7), with an explanation of all bands seen and statement about the reproducibility of the experiments.

The selection of radiolabel sites chosen was based on previous observations of proximity between TFE α and DNA in initiation complexes. DNA to protein cross-linking showed that principal TFE α cross-links were localized to the NT strand between -11 and -9 (Bartlett et al. 2004, Grunberg et al. 2007). Thus, for the protein to DNA cross-link experiments here, -9NT was chosen as a likely positive signal, and -3NT and -4T were chosen as likely negative signals. We added this explanation in the text (line 274-275).

Bartlett, M.S., Thomm, M. & Geiduschek, E.P. Topography of the euryarchaeal transcription initiation complex. *J Biol Chem* **279**, 5894-903 (2004).

Grunberg, S., Bartlett, M.S., Naji, S. & Thomm, M. Transcription factor E is a part of transcription

elongation complexes. *J Biol Chem* **282**, 35482-90 (2007).

Minor Concerns

-line 36 (abstract)-Specify which archaea were used in this study

We added "*Thermococcus kodakarensis (Tko)*". Line 36

-line 55-57- The authors state that archaeal transcription machinery is similar to that found in eukaryotes. Many enzymes in archaea are more similar to their eukaryotic counterparts, so it seems reasonable to remove the clause at the beginning of the sentence.

We edited as the reviewer suggested. Line 55-57

-line 133-134- the statement about TFE α flexibility should not include RNAP because this only becomes apparent later in the text. The sentence should be modified or placed elsewhere in the manuscript.

We modified as the reviewer suggested. Line 129-130

-Provide an SDS-PAGE of purified components. It would also help if the authors provided a UV 260/280 trace and SDS-PAGE for the purification of the complexes used in cryo-EM.

We added SDS-PAGE of apo-RNAP and RNAP-TFE α binary complex in Supplementary Fig. 1a and Supplementary Fig. 2a, respectively. RNAP-TFE α -DNA ternary complex was prepared from RNAP-TFE α binary complex and a synthesized scaffold DNA. RNAP was prepared as previously reported (Jun et al. 2014).

Jun, S.H. et al. The X-ray crystal structure of the euryarchaeal RNA polymerase in an open-clamp configuration. *Nat Commun* **5**, 5132 (2014).

-Consider performing CTF refinement in RELION 3.0/3.1 to improve cryo-EM resolutions/overall quality of maps. This is rather straightforward and usually improves map quality.

We appreciate the reviewer's suggestion and agree that updated versions of RELION and/or recently developed software such as CryoSparrc may improve the resolution of the final map. The procedure is straight forward and should not take much effort.

However, we regret that due to overseas relocation of the author who performed cryo-EM data processing (J Hyun), the access/curation to micrographs and intermediate processing data that are backed up in his previous institute is now difficult to access in timely fashion. Strong travel restriction to/from Korea due to COVID-19 pandemic also prevents direct visit of Dr. Hyun to previous institute for additional data processing.

Regardless, we believe that improvement of the map via further data processing by RELION 3.0/3.1's CTF refinement would be only marginal because the data was collected using a microscope equipped with image spherical aberration (Cs) corrector and hence additional resolution gain from software-mediated aberration correction should not be significant.

Most importantly, we believe that the scientific conclusion drawn in the manuscript will be unaffected by marginal improvements in the cryo-EM maps. More detailed structural investigation would require further extensive data collection using the latest direct electron detector-mediated electron counting movies and image processing with updated software, including the insights into local movements in each domain, but we consider that it could be a new publication of its own right.

-Supplementary figures 1-3-for cryo-EM data, provide angular distribution plot for final maps

We appreciate the reviewer for pointing it out and agree with the suggestion. We have added angular orientation plot for the final cryo-EM map in each of the corresponding supplementary figure.

-Fig 2. State that displayed maps are composite maps in the figure legend

We agree with the suggestion, and additional description of the map shall minimize any confusion. We have clarified the use of composite maps in Fig.2 legend.

-Fig 3D. It would be helpful to distinguish interacting residues residing in the clamp head and stalk from TFE α . Instead of colouring the residues by hydrophobicity, it would help to colour them according to domain/subunit.

We thank the reviewer for the comment and changed coloring the TFE α residues in Fig. 3D as requested (magenta and dark blue for residues interacting with the clamp and stalk domains, respectively).

-Figure 4a- A closer view of the DNA and active site with representative densities would be more helpful than the current rendering. The remainder of RNAP can be left out of this figure.

We thank the reviewer for the suggestion to make a better presentation. Fig. 4a was modified.

-Figure 4c- provide an inset similar to panel b distinguishing which ribbon model is the binary versus ternary complex (grey ternary, coloured binary).

We provided an inset in Fig. 4c as the reviewer suggested.

-Figure 4d- a more focused view of the interactions would help with understanding how the side chains form contacts. Only a portion of RNAP and TFE α need to be shown.

We appreciate the reviewer for the suggestion and modified Fig. 4d as the reviewer suggested.

-Figure 5 boxes- right box- would be helpful if positions of D70 and E72 were indicated

We modified Figure 5 as the reviewer suggested.

-Figure 6a- Provide a colour panel for the relative cross linking efficiencies. A supplementary panel showing how these residues may interact with non-template/template DNA would be helpful.

We appreciate the reviewer for the comment and modified Fig. 6a as the reviewer suggested.

-Methods- For transcription assay, authors use 200 nM TFE α and cross linking assays use 500 nM TFE α . Please provide an explanation for the concentration difference between the assays.

Maximum activity for TFE α in transcription assays is seen with 120 nM or greater. In a previous report, a slight increase in promoter opening was seen when increasing TFE α from 200 to 500 nM, so 500 nM was used in cross-linking experiments to promote maximum occupancy of complexes and best visibility of cross-links.

-Methods- Authors state that to obtain the ternary complex, graphene oxide grids were used to overcome preferential orientation issues. It is unclear if the data from holey carbon grids and the graphene oxide grids were combined. If so, it should be explicitly stated how many particles were used from each dataset.

Indeed, the image data were combined to mitigate the effect of preferred orientation for the 3D reconstruction. We agree that the information is technically important to clarify. We added description of the particles from different dataset in the methods section accordingly. Line 463-465, 510-512, and

519-521

Reviewer #2 (Remarks to the Author):

Review of Jun et al., 2020

General

This is an excellent paper by the Murakami lab that shed light on the structural biology of transcription in archaea, namely the partial X-ray structure of TFE alpha, and the cryoEM structures of the binary RNAP-TFE alpha and ternary RNAP-DNA-TFE alpha complexes. These structures have been sorely missing as they provide a hint for the structural basis of promoter melting and DNA template strand loading into the active site - this makes a substantial contribution to the field, and I think that the article will be of great interest to your readership.

Specifics

Page 4: ‘Our results suggest molecular mechanisms of multiple functions of TFE α during transcription initiation.’

Due to the absence of the essential transcription initiation factors TBP and TFB in the structures determined by the authors, this is an overstatement. The solved RNAP-DNA-TFE alpha structure could just as well represent the RNAP in a conformation that represents the transcription elongation complex (TEC). In this respect its noteworthy that TFE alpha can be incorporated into TECs in vitro (J Biol Chem. 2007 Dec 7;282(49):35482-90. doi: 10.1074/jbc.M707371200).

We appreciate the reviewer’s comment and modified the sentence to tone down the argument. Line 108-110

RNAP-TFE α -DNA ternary structure in this work is missing GTFs TBP and TFB. We could make a model of archaeal OC structure using our structure together with Pol II OC structures as references. As we added in discussion, Pol II adopts the closed clamp state both in the OC and EC structures.

Page 8: ‘[...] indicating that the clamp opening by TFE α is an obligatory step during the PIC formation.’

Transcription initiation is not strictly dependent on TFE, therefore the action ‘by TFE’ cannot be an

obligatory step, it rather happens spontaneously but is enhanced or catalyzed by TFE.

We appreciate the reviewer for the comment and modified the sentence. Line 233-234

Page 9: 'promoter-dependent transcription in transcription complexes containing TFB2 (Fig. 6c)'

This assay has to be explained better. Why TFB2 and not TFB1. And why is TBP not mentioned here?

See also below.

The presence of TBP is specified in Methods, but we have added mention of TBP in the text to help clarify. TFB2 was used in the transcription assay, since the effect of TFE α is greater with TFB2 compared to TFB1 (Micorescu et al. 2008). Line 260-262

Micorescu, M. et al. Archaeal transcription: function of an alternative transcription factor B from *Pyrococcus furiosus*. *J Bacteriol* **190**, 157-67 (2008).

Page 10: 'We suggest a simplified transcription initiation model (Supplementary Movie 2) in archaea based on the model structures of PIC and OC in this study (Fig. 5)'

Without TBP and TFB, the terms PIC and OC can be misleading, it remains an assumption that these complexes represent initiation complexes

Although we tried, we could not determine the cryo-EM structures of complete archaeal PIC and OC. We could make model structures of archaeal PIC and OC using the structures in this work together with Pol II PIC and OC structures. In the simplified PIC and OC models, we focus on the conformations of the clamp and TFE α structures observed in this work. Although TBP and TFB are essential GTFs, to the best of our knowledge, they are not directly involved in the clamp conformation change. We agree with the reviewer that we need the PIC and OC structures including TBP and TFB for the comprehensive understanding of PIC and OC structures.

Page 12: 'The conformational change in the clamp produces a coupled conformation change in the stalk, as observed previously in other RNAP structures^{5,32}.' The article ends a bit abrupt, and the topic would benefit from additional elaboration. The authors should also include a final paragraph here to reflect on the structural differences between initiation and elongation complexes. I'm happy to go along with that the RNAP-TFE-DNA complexes presented in this article are resembling initiation complexes, but without TBP/TFB they cannot strictly speaking be called just that. The results in the paper are excellent - a discussion of this topic would suffice to alleviate my concerns. Katsu published the first archaeal RNAP-Spt4/5 cryoEM structure in PNAS years ago.

- What are the similarities and differences between the two structural states of RNAP?
- What conformational changes in RNAP have to occur during promoter escape?

- How does the RNAP-TFE-DNA structure support the notion that TFE can associate with TECs?

We appreciate the reviewer for the suggestion and added a paragraph as the reviewer suggested. Line 338-345

The RNAP clamp mobility and positioning has been characterised in solution using smFRET, as cited earlier on in the manuscript. But differences remain, and it would be good if the authors could compare their new structural information on the background of the clamp opening and closure described in solution through the transcription cycle.

We appreciate the suggestion and added a paragraph as the reviewer suggested. Line 346-353

Finally, detailed models of promoter melting and DNA template strand loading have been proposed for the simpler bacterial RNAPs. The manuscript would benefit from a detailed comparison of these processes between the two systems. Bacteria do not use TFE-like factors but some, e.g. the Mtb RNAP, uses analogous factors such as CarD with interesting differences and commonalities.

We appreciate the reviewer for the suggestion to make better paper and added a paragraph as the reviewer suggested. Line 354-366

Methods.

Page 17: ‘In vitro transcription and cross-linking. Mutant TFE variants containing site-specific Bpa substitutions were made using the pSUP system as described^{28,46}. Transcription reactions were conducted using recombinant TBP, TFB2, and TFE and native RNAP, assembled at the *gdh* promoter as described previously, with RNAP at 10 mM, TBP at 60 nM, TFB2 at 120 nM, and TFE at 200 nM⁴⁷. TFE Bpa cross-linking in preinitiation complexes was tested using 1 nM site-specifically radiolabeled *gdh* promoter DNA, at 10 nM RNAP, 60 nM TBP, 120 nM TFB1, and 500 nM TFE^{alpha}. Cross-links were formed and products analyzed as described previously⁴⁶.’

Do you really mean 10 millimolar RNAP? Probably 10 micromolar, or was it nanomolar?

Why use TFB1 in the cross-linking assay, and not TFB2 as in the transcription assay? Not that this is a major problem, but please explain.

We thank the reviewer and changed the error. It's 10 nM RNAP. Line 549

TFB1 was used in cross-linking assays to more closely replicate the conditions in which DNA (-9NT) to protein TFE cross-links were first observed and identified (Bartlett et al. 2004, Grunberg et al 2007, Micorescu et al. 2008).

Bartlett, M.S., Thomm, M. & Geiduschek, E.P. Topography of the euryarchaeal transcription initiation complex. *J Biol Chem* **279**, 5894-903 (2004).

Grunberg, S., Bartlett, M.S., Naji, S. & Thomm, M. Transcription factor E is a part of transcription elongation complexes. *J Biol Chem* **282**, 35482-90 (2007).

Micorescu, M. et al. Archaeal transcription: function of an alternative transcription factor B from *Pyrococcus furiosus*. *J Bacteriol* **190**, 157-67 (2008).

This is a nice piece work, well done Katsu & team!

Finn Werner

REVIEWERS' COMMENTS

Reviewer #1 (Remarks to the Author):

The authors have addressed all reviewer concerns. The authors should be congratulated for their efforts and important contribution to the field.